# CoolPrompt: An Automatic Prompt Optimization Framework for Large Language Models

## Abstract

The effectiveness of Large Language Models (LLMs) is highly dependent on the design of input prompts. Manual prompt engineering requires a domain expertise and prompting techniques knowledge that leads to a complex, time-consuming, subjective, and often suboptimal process. We introduce CoolPrompt as a novel framework for automatic prompt optimization. It provides a complete zero-configuration workflow, which includes automatic task and metric selection, also splits the input dataset or generates synthetic data when annotations are missing, and final feedback collection of prompt optimization results. Our framework provides three new prompt optimization algorithms ReflectivePrompt and DistillPrompt that have demonstrated effectiveness compared to similar optimization algorithms, and a flexible meta-prompting approach called HyPE for rapid optimization. Competitive and experimental results demonstrate the effectiveness of CoolPrompt over other solutions.

## 1 Introduction

Large Language Models (LLMs) such as GPT-4 (Achiam et al., 2023), Claude AI (Anthropic, 2024), DeepSeek (Liu et al., 2024), Grok (xAI, 2025), and LLaMA (Touvron et al., 2023) have revolutionized artificial intelligence transitioning from task-specific solutions to general-purpose foundation models (Vivekananda et al., 2025; Yang et al., 2023b) and driving their rapid adoption across research and industry (Zhao et al., 2024). They have exhibited unprecedented effectiveness due to their remarkable performance in natural language understanding (Karanikolas et al., 2023), text generation (Brown et al., 2020; Bao et al., 2023), code generation (Chen et al., 2021; Zhang et al., 2024a), and reasoning (Lewkowycz et al., 2022). Meanwhile their operational efficacy is fundamentally mediated by the quality of prompt design (Liu et al., 2023), where prompts serve as computational directives from human to model (Kadavath et al., 2022).

Prompt engineering is the practice of designing input instructions to elicit desired model behavior and has emerged as a critical and rapidly evolving discipline (Vatsal & Dubey, 2024; Schulhoff et al., 2024). The process involves creating prompts, which can include questions, instructions, or templates that use the embedded knowledge of the model to maximize performance on tasks. Unlike traditional fine-tuning, prompt engineering does not require modifying the model's weights; instead, it leverages the model as a fixed generalist 'language computer'. Prompt engineering methods range from simple input templates such as few-shot techniques (Wang et al., 2020) to advanced strategies such as Chain-of-Thought prompting (Wei et al., 2022), Self-Discover (Zhou et al., 2024), Tree-of-Thoughts (Yao et al., 2023a), ReAct (Yao et al., 2023b) and etc (Vatsal & Dubey, 2024). Moreover, recent advances have enabled LLMs to self-generate and iteratively refine their own prompts through in-context learning and reinforcement signals, automating aspects of the prompt design process (Li et al., 2025c).

Manual prompt engineering remains fraught with challenges that limit its potential, performance, scalability, and accessibility. First, designing high-performance prompts typically requires extensive trial-and-error, deep domain expertise, and prompting techniques knowledge; therefore the process is often time-consuming and nonsystematic. Second, although current LLMs are trained in human-generated text data, the effectiveness of prompt generation is also influenced by factors such as input and output format (Min et al., 2022), placement of few-shot examples (Lu et al., 2022), the use of key trigger words and tokens (Xie et al., 2022; Shin et al., 2020), and the elimination of redundant

tokens and words (Wang et al., 2025). Consequently, these factors reduce the relative importance of semantic clarity in human-oriented content and narratives, thus slowing down the process of manual prompt design. Moreover, prompt effectiveness often exhibits poor transferability across tasks, datasets, and even different LLM architectures (Zhang et al., 2024b; Su et al., 2022; Zhou et al., 2022), undermining reproducibility and scalability and requiring additional time and resources to refine prompts.

One of the most profound advances enabled by LLMs is the development of automatic prompt optimization (autoprompting) that includes different algorithms and optimization strategies to automate the design, selection, and refinement of prompts supplied to language models (Li et al., 2025a). Autoprompting leverages methods such as llm-based and planning approaches (Zhou et al., 2022; Yang et al., 2024), reinforcement learning (Wang et al., 2023a; Deng et al., 2022; Zhang et al., 2023), evolutionary algorithms (Wang et al., 2025; Singh et al., 2022; Guo et al., 2023), and meta-optimization (Yang et al., 2023a; Singh et al., 2022; Pryzant et al., 2023) to optimize prompts. Studies show that automatic prompt optimization can achieve higher efficiency, consistency and scalability even with manual prompting by experts (Zhou et al., 2022). It reduces human workload while improving the robustness between tasks and generalization of prompt strategies.

Despite these advances, current autoprompting methods still have several drawbacks. First, rapid efficacy system evaluation requires comprehensive evaluation methodologies incorporating specialized testing frameworks, domain-adapted performance metrics, and statistically significant experimental designs, collectively imposing substantial computational and temporal resource requirements (Chang et al., 2024). Second, the stage of prompt engineering remains costly, as there is no intuition or universal methods and strategies in selecting prompting and autoprompting methods, and the complexity of problem evaluation for specific data creates a barrier to prompt engineering (Li et al., 2025a; Vatsal & Dubey, 2024). Third, many current autoprompting implementations are tailored to proprietary LLMs, making it difficult to use custom or open-source models for specific tasks, which reduces the democratization of LLM selection and usage (Zhou et al., 2022; Guo et al., 2023). Finally, conventional prompt engineering approaches exhibit limited generalizability in various task domains and applications (Chang et al., 2024).

To address these fundamental limitations, we introduce **CoolPrompt**, a comprehensive automatic prompt optimization framework that serves as an alternative to manual prompt design, providing a complete workflow from task definition to prompt evaluation. This framework offers a quick start to prompt optimization with zero expertise and minimal prompt engineering requirements. **CoolPrompt** includes automatic task and metric selection for task assessment, splitting the input dataset or generating synthetic data when annotations are missing, and final feedback collection of prompt optimization results. Our framework includes two new innovative autoprompting algorithms: ReflectivePrompt and DistillPrompt that have demonstrated effectiveness compared to similar solutions and a flexible meta-prompting approach called HyPE for rapid optimization.

**CoolPrompt** allows machine learning and prompt engineers, researchers, and practitioners to take advantage of state-of-the-art prompt-based optimization without requiring deep knowledge of the inner workings of LLMs or optimization algorithms (Baclic et al., 2020). Beyond its technical contributions, this work addresses the main challenges of ensuring accessibility when deploying LLMs, removing expert barriers, and offering intuitive interfaces. This standardization is a key to democratizing and accelerating the adoption of LLMs in industries and research areas where operational engineering knowledge is limited but the potential for application is high.

The present work contributions are the following:

1. We present a zero-configuration framework that advances a wide range of LLMs and automates the full prompt optimization pipeline as an alternative manual prompting design, from task definition to automatic prompt evaluation.

2. We propose a synthetic data generation approach that eliminates data bottlenecks in prompt optimization.

3. We propose several optimization strategies for short-term optimizations: meta-prompting approach HyPE, and for long-term optimizations: autoprompting algorithms ReflectivePrompt and DistillPrompt.

The experimental studies show that CoolPrompt achieves competitive performance on different tasks such as mathematical reasoning, question answering, classification, summarization, and natural language understanding. Its cost-aware optimization further allows users to tailor performance-efficiency trade-offs, validating both its practical utility and generalizability.

## 2 RELATED WORK

### 2.1 PROMPTING TECHNIQUES

Recent developments in prompt engineering have shown significant advances in prompt design techniques (Liu et al., 2023). For example, Few-shot prompting (Wang et al., 2020) provides instructive examples to guide the model. Chain-of-Thought prompting (Wei et al., 2022) proved an effectiveness by generating the model's reasoning process before generating a final answer. Motivated on this, more advanced reasoning prompt designs have emerged. Self-Discover (Zhou et al., 2024) selects pre-existing reasoning chains, adapts them to the specific task, and applies them directly. Self-Consistency (Zhou et al., 2024) samples multiple reasoning paths and implements them to produce the most consistent answer. Tree-of-Thoughts (Yao et al., 2023a) and Graph-of-Thoughts (Besta et al., 2024) generate various decomposed reasoning variations, which are then evaluated and selected, thus increasing the depth of the exploration. ReAct (Yao et al., 2023b) goes further by generating reasoning that translates into actions, while reflecting on previous steps, it is commonly integrated within Retrieval-Augmented Generation (RAG) (Yao et al., 2023b) pipelines and agent-based systems.

Recent research has also revealed self-critique methods to minimize risks of hallucinations such as Chain-of-Verification (Dhuliawala et al., 2023) and Self-Refine (Madaan et al., 2023), as well as agentic prompting frameworks that empower LLMs to operate autonomously with tool-use capabilities. In addition, multimodal prompting techniques have extended prompt engineering beyond text to include image (Hakimov & Schlangen, 2023; Oppenlaender, 2024), audio (Wang et al., 2024), video (Brooks et al., 2024), and segmentation prompting (Tang et al., 2025).

### 2.2 AUTOMATIC PROMPTING ALGORITHMS

Currently, a variety of auto-prompting algorithms have been developed, based on different optimization methods. Specifically, EvoPrompt (Guo et al., 2023) and PromptBreeder (Fernando et al., 2024) employ an evolutionary approach, where a large language model (LLM) serves as a selection, mutation, or recombination operator. PromptAgent (Wang et al., 2023b) and StablePrompt (Kwon et al., 2024) utilize Reinforcement Learning (RL), optimizing prompts using a reward model. Solutions such as iPrompt (Singh et al., 2022) and OPRO (Hong et al., 2024) are built on LLMs or foundation models (FMs), leveraging meta-prompts to modify the optimization pipeline. The primary motivation for exploring autoprompting comes from research on the Automatic Prompt Engineer (Zhou et al., 2022), where it was demonstrated that modern LLMs can handle prompt generation and optimization tasks comparable to or even better than human experts.

### 2.3 PROMPT OPTIMIZATION LIBRARIES

Current prompt optimization solutions offer a variety of functionalities and optimization modules. AdalFlow (Yin & Wang, 2025) provides an auto-differentiable framework that supports both zero-shot and few-shot prompt optimization, along with rapid construction of LLM, RAG, and Agent pipelines. PromptWizard (Agarwal et al., 2024) enables automatic prompt optimization through prompt refinement and synthetic data generation. PromptoMatrix (Murthy et al., 2025) showcases an end-to-end prompt optimization pipeline that employs multiple strategies and evaluates performance on synthetic data.

## 3 PROPOSED FRAMEWORK

### 3.1 ARCHITECTURE OVERVIEW

CoolPrompt is a comprehensive framework, featuring a complete pipeline for automated prompt optimization. The system is designed for both direct usage and seamless integration with other platforms and systems. The complete framework architecture and user workflow are presented in Fig. 1. See Appendix A.5 for key features details.

The architecture comprises several core functional modules:

1. **PromptTuner** is a primary interface class for parameter configuration and optimization pipeline execution.

2. **Evaluator** is a module for assessing prompt performance in datasets, incorporating multiple metrics for both classification and generation tasks.

3. **PromptOptimizer** is a versatile optimization module that supports short-term adaptations through prompt engineering techniques with meta-prompts and long-term automatic prompt optimization algorithms.

4. **PromptAssistant** is a LLM-based component with predefined meta-prompts to interpret prompt optimization results for users.

5. **Synthetic Data Generator** is an auxiliary module for synthetic data generation when no input dataset is provided.

6. **Task Detector** is an automated task classification component for scenarios without explicit user-defined task specifications.

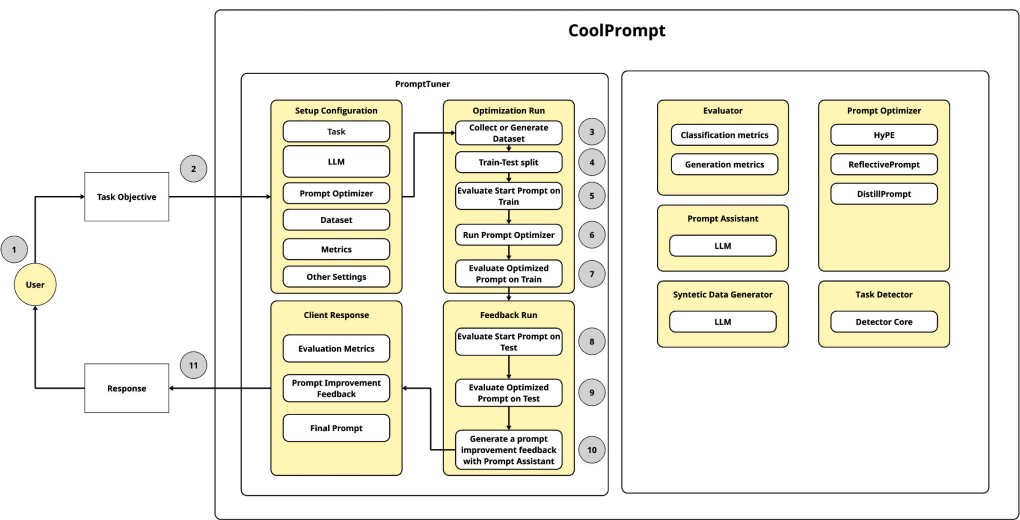

Figure 1: CoolPrompt system architecture and user workflow

### 3.2 OPTIMIZATION WORKFLOW

Fig. 1 illustrates the sequential workflow steps of CoolPrompt. During step 1, the user provides a start prompt for optimization initialization. Additionally, users may explicitly specify the following parameters: a task type (classification or generation), an evaluation metric, an optimization method, LLMs supported via LangChain integration for the target and assistant LLM, a problem description for ReflectivePrompt, a labeled dataset with target responses, and a train-test split ratio.

In Step 2, the system employs automated fallback mechanisms to handle unspecified parameters. This includes using a Task Detector for automatic task classification, predefined metrics for evaluation, default LLM interfaces, and auto-generated problem descriptions. HyPE serves as the default method in PromptOptimizer, and predefined split ratios are applied for dataset partitioning.

Steps 3-4 perform dataset validation, using synthetic dataset generation with corresponding labels via the Synthetic Data Generator when it is required, followed by train-test sampling. Step 5 evaluates the initial prompt on the training subset. Step 6 executes prompt optimization through the selected PromptOptimizer method. Step 7 assesses the final optimized prompt on test set.

The comparative evaluation between the initial and optimized prompts occurs during Steps 8-9. Step 10 employs the PromptAssistant component to generate self-improvement feedback by analyzing initial and final prompt performance. The workflow finishes at Step 11 with a delivery of pipeline results: comprehensive train-test evaluations with metrics and actionable prompt refinement feedback.

## 3.3 PROMPT OPTIMIZATION METHODS

### 3.3.1 HYPE

HyPE (Hypothetical Prompt Enhancer) is a rapid meta-prompting approach for adaptive prompt enhancement that asks a large language model to generate a hypothetical instructive prompt which solves the same underlying task as the user's query. The design intentionally avoids multi-round prompt search or ensembles of hand-crafted transformation rules: instead, HyPE exploits the model's internal knowledge of effective prompting patterns to produce an immediately usable reformulation in one extra forward pass, giving lightweight, task-adaptive prompt optimization with minimal engineering.

The idea of HyPE is motivated by the HyDE method (Gao et al., 2023), which synthesizes a hypothetical document to improve the retrieval process. HyPE applies the same idea to prompt formulation rather than retrieval. The meta-prompt template and our procedure for selecting it are described in Appendix A.2.3; that analysis shows that templates asking for a concise, self-contained instruction with an explicit I/O specification yield the most consistent gains.

Building on this concept, HyPE stands out from previous prompt optimization techniques. Methods like chain-of-thought prompting require task-specific exemplars, while automated strategies often depend on multi-step LLM calls or rule-based transformations, incurring substantial computational or engineering overhead. In contrast, HyPE's single-step generation leverages the model's inherent, pretrained understanding of instructional language. This intrinsic efficiency yields prompts that are both more generalizable and precise than the original query, effectively distilling the task's core requirements into an optimal instruction for the model itself, without the need for external search or exemplars.

### 3.3.2 REFLECTIVEPROMPT

ReflectivePrompt is an evolutionary-based prompt optimization method, which is built on the idea of Reflective Evolution (Ye et al., 2024). Using the concepts of textual gradient (Li et al., 2025b) and self-reflection (Zhao et al., 2025), it provides remarkable results in different areas of autoprompting tasks. All the data required to run the method: a dataset, a description of the target problem, and an initial user prompt. The remaining individuals of the first population are created by producing diverse paraphrases of the user prompt.

A key feature of ReflectivePrompt is delegating a decision on the specifics of mutation to the model itself. The workflow of each epoch of the ReflectivePrompt algorithm is shown in Fig. 2.

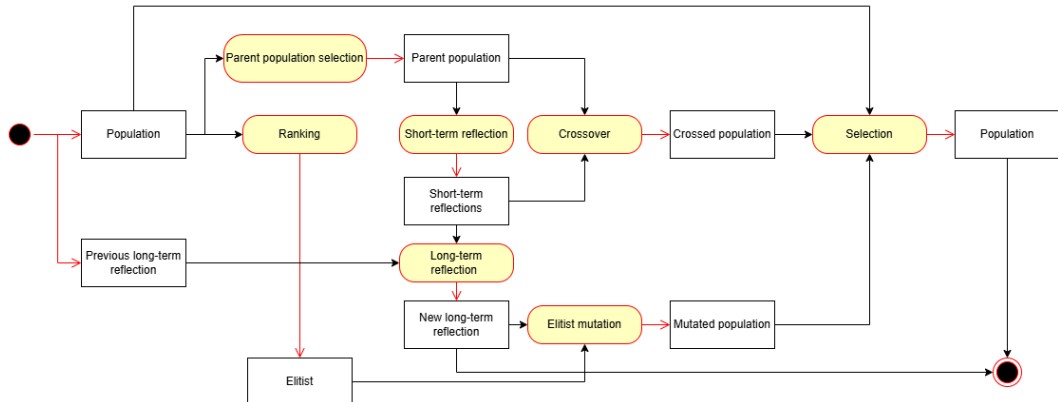

Figure 2: The reflective evolution pipeline in ReflectivePrompt. First, given the initial population, the parent population is sampled. Second, current short-term reflections are produced and crossover operation takes its place. Third, using current short-term reflections, long-term reflection updates and elitist-based mutation generates new individuals. After all, new prompts are evaluated and selected for new epoch.

ReflectivePrompt implements two evolutionary operators: crossover and elitist mutation. They both leverage short-term and long-term reflections to improve their effectiveness. Crossover creates a new prompt from two parent individuals using generated short-term reflection. Initially, a set of parent pairs is sampled according to these rules:

1. Each prompt is selected with probability proportional to its fitness score.
2. One prompt can be selected in multiple parent pairs.
3. Within each pair, one prompt must have a strictly higher score than the other.

The difference in fitness scores is required to determine the superior and inferior prompts in each parent pair, since the short-term reflection is focused on identifying qualities and dissimilarities that yield higher-scoring prompts. Short-term reflection consists of the model generating reflective analyses and hints that are then used to achieve better crossover offspring. In summary, the set of short-term reflections constitutes an analytics of the individuals in the current population.

The elitist mutation operator generates new individuals using the best prompt in the current population and long-term reflection. This operator enables local search in the area of the present optimum. Long-term reflection is updated in each epoch based on its prior version and all short-term reflections produced in that generation. It contains a distilled summary of the model reasoning about the current population and accumulates knowledge across all epochs of the evolution by incorporating its previous state.

The experiment results are provided in the Appendix A.3.1.

### 3.3.3 DISTILLPROMPT

DistillPrompt is a gradient-free automatic prompt optimization algorithm based on iterative prompt distillation. The method employs prompt compression, semantic reformulation, and dynamic example integration to enhance prompt effectiveness across diverse NLP tasks.

This method is based on the idea of the Tree-of-Thoughts prompting technique. At each epoch, DistillPrompt uses the best prompt from the previous iteration according to the target metric. For the first epoch, the initial user prompt is employed.

DistillPrompt workflow is illustrated in Fig. 3. The pipeline is the following. First, several variations of the initial prompt are generated. These diverse modifications are used to analyze the search space from different perspectives. The generated variations explore the search area in mostly blind and inefficient way, and in order to cope with this, the second step incorporates knowledge embedding.

The objective is to specialize the prompt for the task while preserving its original formulation as much as possible. To achieve this, several examples are randomly sampled from the training dataset and provided to the model to extract some key principles and ideas that are necessary for solving these training examples. The created concepts are then embedded into the prompt.

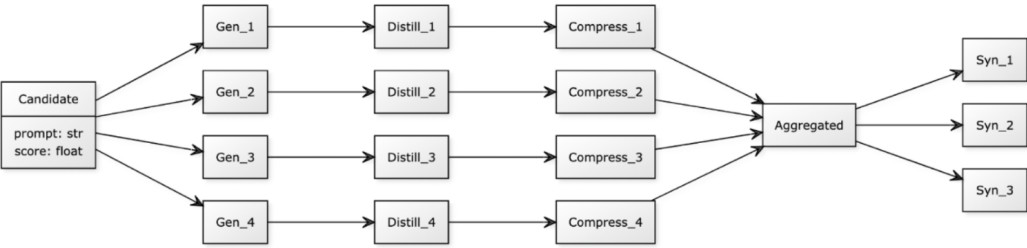

Figure 3: DistillPrompt workflow. First step - generating paraphrased variants (*Gen_i*). Then prompts are embedded with training data knowledge (*Distill_i*). Third step is compressing promtps into several sentences (*Compress_i*). Final two steps are aggregation (*Aggregated*) and creating final diverse variants (*Syn_i*).

However, there is a risk of the model "overfitting" to the given examples and embedding the provided questions and labels itself rather than generalizing for the whole task. To mitigate this, the next step involves instruction compression. The LLM reformulates each prompt into a small number of sentences, preserving the core content of both the original formulations and the embedded task-solving principles.

Since the examples from training data were sampled independently and randomly for each candidate, the resulting insights may vary. Thus, the natural progression is to merge the compressed candidates into a single distilled prompt accumulating the collective ideas. The final stage generates the variations of aggregated prompt, similar to the very first step, and then the new best prompt is selected from these newly created candidates.

The experimental results are shown in the Appendix A.4.1

## 4 EXPERIMENTAL EVALUATION

### 4.1 EXPERIMENTAL SETUP

**Baselines** We select the following popular automatic prompt optimization frameworks: Promptomatix (Murthy et al., 2025), AdalFlow (Yin & Wang, 2025), and Promptify (Pal, 2022)), and manual zero-shot prompts for each task, also as a starting point for prompt optimization.

**LLM** As other automatic prompt optimization frameworks have limitations of usage only proprietary LLMs, we select the gpt-3.5-turbo model (Brown et al., 2020) with specific generation parameters, mentioned in Appendix A.1.

**Datasets** We select five benchmark datasets: SQuAD_2 (Rajpurkar et al., 2018) for a question answering, GSM8K (Cobbe et al., 2021) for a mathematical reasoning, CommonGen (Lin et al., 2020) for a natural language understanding, XSum (Narayan et al., 2018) for a summarization, and AG News (Zhang et al., 2015) for a text classification.

**Evaluation Metrics** We select metrics for each dataset according to their task specificity: BERTScore (Zhang* et al., 2020) for SQuAD, CommonGen and XSum to evaluate semantic and n-gram correctness between model responses and target answers; EM (Exact Match) for GSM8K to match exact target mathematic result; F1 Macro for AG News to evaluate classification accuracy according to class imbalance.

**Experiment Details** Please see Appendix A.1 for details.

## 4.2 EVALUATION AND COMPETITIVE ANALYSIS RESULTS

Table 1 presents the average results across all runs between CoolPrompt optimization methods and other automatic prompt optimization frameworks.

Table 1: Comparative analysis between autoprompting frameworks (central group) and CoolPrompt optimization methods (right group). **Bold** values indicate results that outperform others.

| Dataset | Metric | Manual Zero-shot Prompt | Promptify | AdalFlow | Promptomatix | CoolPrompt ReflectivePrompt | CoolPrompt DistillPrompt | CoolPrompt HyPE |
|---|---|---|---|---|---|---|---|---|
| SQuAD_2 | BertScore | 0.875 | 0.905 | 0.920 | 0.918 | **0.934** | 0.922 | 0.930 |
| GSM8K | EM | 0.527 | 0.615 | **0.753** | 0.728 | 0.732 | 0.722 | 0.710 |
| CommonGen | BertScore | 0.871 | 0.885 | 0.904 | 0.902 | **0.913** | 0.911 | 0.907 |
| AG News | F1 | 0.705 | 0.841 | 0.722 | **0.858** | **0.858** | 0.845 | 0.791 |
| XSum | BertScore | 0.823 | 0.233 | 0.841 | 0.857 | **0.872** | 0.842 | 0.851 |

Table 2 presents a comprehensive feature comparison between CoolPrompt and other frameworks, which includes six main key features for an automatic prompt optimization process: Auto Data, Auto Task, Custom Model Usage, Zero Config, Auto Metric, Optimization Feedback.

Table 2: Feature comparison between automatic optimization libraries. Attention was paid to the following key features: automated generated dataset (**Auto Data**), automatic task determination (**Auto Task**), the ability to work with open-source and custom LLMs (**Custom Model Usage**), the ability to use only start prompt for optimization (**Zero-Config**), automatic metric determination (**Auto Metric**), interpretation of prompt optimization results (**Optimization Feedback**).

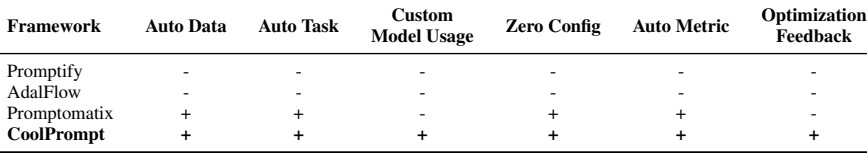

| Framework | Auto Data | Auto Task | Custom Model Usage | Zero Config | Auto Metric | Optimization Feedback |
|---|---|---|---|---|---|---|
| Promptify | - | - | - | - | - | - |
| AdalFlow | - | - | - | - | - | - |
| Promptomatix | + | + | - | + | + | - |
| **CoolPrompt** | + | + | + | + | + | + |

## 5 DISCUSSION AND LIMITATIONS

According results presented in Table 1, CoolPrompt demonstrated competitive performance efficiency on the majority of benchmark tasks. On these datasets the results obtained using CoolPrompt are comparable to those of other frameworks. For generative tasks, the prompts generated by Cool-Prompt yielded superior results compared to other libraries.

The competitive analysis in Table 2 indicates that the most similar framework is Promptomatix, which is distinguished by its more limited options for selecting custom models compared to Cool-Prompt. It is also worth mentioning the implemented criterion for the automatic selection of evaluation metrics. The current version of CoolPrompt supports the selection of various metrics, depending on previously chosen or automatically detected task type, with F1 and BertScore selected by default as the most representative metrics for providing a balanced assessment between model responses and target outputs.

CoolPrompt still has several limitations. First of all, the current library implementation is limited to the textual modality. Moreover, we measure LLM responses only with a standard set of evaluation metrics: for classification tasks (accuracy, recall, precision, F1); for text generation tasks (BLEU, ROUGE, METEOR (Banerjee & Lavie, 2005), BERTScore, ExactMatch). Besides from that, when operating within highly specialized domains (e.g., medicine, jurisprudence, biology), the appropriate language model is usually required to be used, while datasets considered in our research for experiments cover common domain area.

## 6 CONCLUSION

In this work, we have shown CoolPrompt a zero-configuration framework that automates the full prompt optimization pipeline as an alternative manual prompting design, demonstrating competitive effectiveness across diverse tasks. CoolPrompt represents a significant advancement in the field of automatic prompt optimization.

The principles embedded within automation, efficiency, and accessibility it as a key tool for the next generation of LLM-based applications. CoolPrompt plays a particularly important role in the context of the continuous evolution of language models, fostering broader participation in AI development and accelerating the adoption of these technologies across various subject domains and user communities by removing barriers in prompt design.

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

# A  APPENDIX

## A.1  EXPERIMENT DETAILS

For each task in the evaluation experiment, we provided a training and validation data split of 30 samples from the original dataset with a train split ratio of 0.2. We chose the usage of original dataset because not every automatic optimization framework is capable for the generating syntetic data. See Appendix A.2.1 for the ablation study on the quality of the generated dataset compared to the original.

We evaluated each method in the comparison in 3 runs in order to obtain more objective results due to probabilistic LLM output generation, where Table 1 presents the average results across all runs. Prompt optimization methods DistillPrompt and ReflectivePrompt were run with a number of epochs of 5 and for the second method the prompt population size of 10. LLM run with the following generation parameters: temperature was 0.7 and maximum number of new tokens was 3500.

**Temperature** controls the randomness of the generated text, where low temperatures produce deterministic text and high temperatures foster greater creativity and diversity. We set temperature to 0.7 to ensure a variety of LLM responses and optimization runs.

**Maximum number of new tokens** constrains the limitation a size of generated tokens. In order to avoid overly restricting the model, we set this limit to 3500 tokens.

## A.2  ABLATION STUDIES

### A.2.1  GENERATED SYNTHETIC DATA QUALITY

Synthetic data generation represents a key feature for automatic prompt optimization in the cases of lack or absence of the dataset that corresponds to the given problem. This is the reason why it is crucial to understand whether the generated data can equally and effectively replace the real one in terms of autoprompting optimization.

For this experiment, we used open-source popular models: ministral-8b-instruct-2410 and qwen3-4b-instruct-2507. The experiment was conducted on the same datasets as in the main part of the paper. Firstly, for each task we provided a training and validation data split of 40 samples from the original dataset with a split ratio of 0.5 and ran the optimization according to it. Secondly, we used only the initial prompt to automatically generate the data. Finally, obtaining the best prompts for each approach, we evaluated them on the entire original dataset, and the resulting metrics are shown in Table 3.

Table 3: Comparison between the ReflectivePrompt metrics with real data and synthetic data usage. Synthetically generated dataset has proven to be the good replacement when no real-word data can be used. **Bold** values indicate results that outperform others.

| Dataset | Metric | Model | Real data | Synthetic data |
|---------|--------|-------|-----------|----------------|
| SQuAD_2 | BertScore | ministral | **0.967** | **0.967** |
| SQuAD_2 | BertScore | qwen3 | **0.997** | 0.905 |
| GSM8K | EM | ministral | **0.779** | 0.741 |
| GSM8K | EM | qwen3 | **0.882** | 0.726 |
| CommonGen | BertScore | ministral | **0.918** | **0.918** |
| CommonGen | BertScore | qwen3 | **0.918** | 0.912 |
| AG News | F1 | ministral | 0.256 | **0.264** |
| AG News | F1 | qwen3 | **0.655** | 0.646 |
| XSum | BertScore | ministral | **0.860** | 0.773 |
| XSum | BertScore | qwen3 | 0.852 | **0.855** |

The results of this experiment show that in half of the cases, synthetic data generation proved to be even a better replacement for real-world data. The quality of the generated data strictly relies on the

general abilities and knowledge of the model, so we can see that the larger gap between results can sometimes occur on qwen3 model with fewer parameters compared to ministral.

### A.2.2 GENERATING SYNTHETIC DATASET WITH MORE POWERFUL MODEL

This experiment was held to determine whether the synthetic data that was generated by a more powerful model (in our case, we used gpt-3.5-turbo) can outperform the generation by the model itself. The comparison was done between gpt-based and ministral-based generated data. All optimization processes were performed using ministral-8b-instruct model (GPT model was only used to generate data). The metrics in Table 4 show that generating data from a larger model can sometimes lead to significant improvements, but in our particular example it achieves lower average score.

Table 4: ReflectivePrompt metrics in two use cases: when data is generated by ministral itself and when data was generated by GPT-3.5-turbo and provided for ministral language model. **Bold** values indicate results that outperform others.

| Dataset | Metric | Ministral-generated data | GPT-generated data |
| --- | --- | --- | --- |
| SQuAD_2 | BertScore | **0.967** | 0.756 |
| GSM8K | EM | 0.741 | **0.813** |
| CommonGen | BertScore | **0.918** | 0.918 |
| AG News | F1 | **0.264** | 0.209 |
| XSum | BertScore | 0.773 | **0.853** |

### A.2.3 SELECTING THE META-PROMPT FOR HYPE

HyPE relies on a single meta-prompt to guide its entire prompt optimization process, so the meta-prompt's quality is critical to the quality of generated prompts. Our initial approach was to design a meta-prompt that directly reflects the method's purposes (see Fig. 4).

---

Please write a hypothetical instructive prompt for the following query to make a large language model answer the question.
Query: {QUERY}
Prompt:

---

Figure 4: Initial meta-prompt for HyPE

To evaluate this initial meta-prompt we measured HyPE's behavior using task-specific prompt templates for classification and generation tasks (see Fig. 5 and 6). The templates, shown below, were designed to structure the LLM's final answer. In these templates, INPUT denotes the instance from the dataset (e.g., a news article to be classified), and LABELS denotes the list of dataset's classification labels (for example, the AG News label set).

```
{PROMPT}
Answer using the label from [{LABELS}].
Generate the final answer bracketed with <ans> and </ans>.
Examples:
1. Labels are [(A), (B), (C)] and you chose the first option
   Output will be: <ans>(A)</ans>
2. Labels are [A, B, C] and you chose the first option
   Output will be: <ans>A</ans>
Input: {INPUT}
Response:
```

Figure 5: Prompt template for classification task

```
{PROMPT}
Provide a direct answer without additional explanations or commentary.
Generate the final answer bracketed with <ans> and </ans>.
INPUT: {INPUT}
RESPONSE:
```

Figure 6: Prompt template for generation tasks

Running HyPE with the initial meta-prompt and inspecting the generated hypothetical instructive prompts revealed several systematic failure modes that limited downstream performance and robustness:

1. The model sometimes directly answered the user's query instead of producing a hypothetical instructive prompt that would instruct another model to solve the task.
2. Parts of the model's internal reasoning occasionally leaked into the generated prompt, producing noisy answers.
3. When the original query was underspecified or overly general, the generated hypothetical prompt injected extraneous details, altering the task's domain. For example, basic prompt for GSM8k "Given a context answer on the question." was augmented as "In 200 words, provide a detailed explanation of the concept of supply and demand in economics...", though the task's domain is math solving.
4. Special formatting, code fragments and placeholders present in the original query were not reliably preserved.
5. The language of the hypothetical prompt did not always matched the language of the original query.

To address these limitations we revised the meta-prompt to impose stricter, explicit constraints on the form and content of the generated hypothetical prompt. The key design changes and refinements were:

1. Emphasize that the hypothetical instructive prompt *must* solve the same underlying task as the original query and it should not directly answer the query.
2. Require a strict output format (using explicit tags) to ensure clean parsing and prevent reasoning leakage.
3. We incorporated using of an auxiliary `problem_description` field during generation: HyPE will use a short problem description to focus the hypothetical prompt. If the user does not provide `problem_description`, Synthetic Data Generator generates one before the optimization.
4. Add hard constraints to preserve the original language, any special formatting, and code blocks precisely.

The resulting final meta-prompt integrates these refinements into a comprehensive instruction set so that generated hypothetical prompts are consistently instructive, task-faithful, and machine-parseable (shown in Fig. 7).

---

You are an expert prompt engineer. Your only task is to generate a hypothetical instructive prompt that would help a large language model effectively answer the following query. The prompt must solve the same underlying task as the original query while being more effective.
### HARD CONSTRAINTS ###
1. LANGUAGE:
   - Output MUST be in the EXACT SAME LANGUAGE as the query.
2. CONTENT:
   - Output ONLY the hypothetical instructive prompt - do NOT answer the original query directly.
   - The hypothetical prompt must solve the same task as the oiginal query provided by user.
   - If the original query contains any code snippets, you must include it in final prompt.
3. TECHNICAL PRESERVATION:
   - Code blocks must be preserved with original syntax and formatting.
   - Variables, placeholders ({{var}}), and technical terms kept unchanged.
   - Markdown and special formatting replicated precisely.
### YOUR OUTPUT FORMAT ###
[PROMPT_START]<your hypothetical instructive prompt here>[PROMPT_END]
### INPUT ###
User's query: {QUERY}
Problem description: {PROBLEM_DESCRIPTION}
### OUTPUT ###
Hypothetical Instructive Prompt:

---

Figure 7: Final meta-prompt for HyPE

We evaluated the performance of the initial and final meta-prompts across several benchmarks using gpt-3.5-turbo with temperature 0.7 and maximum 3500 new tokens. As shown in Table 5, the final meta-prompt yields a substantial improvement across all datasets and metrics, confirming the that explicit constraints on content, format and preservation materially improve HyPE's optimization quality.

Table 5: GPT-based HyPE optimization results comparison between meta-prompts. **Bold** values indicate results that outperform others.

| Dataset | Metric | Initial meta-prompt | Final meta-prompt |
|---|---|---|---|
| SQuAD_2 | BertScore | 0.917 | **0.935** |
| GSM8K | EM | 0.260 | **0.732** |
| CommonGen | BertScore | 0.866 | **0.909** |
| AG News | F1 | 0.691 | **0.781** |
| XSum | BertScore | 0.767 | **0.861** |

In addition to the structural refinements described above, we conducted a controlled study to assess how different phrasings of the target instruction inside the meta-prompt affect HyPE's optimization performance. Specifically, we compared four variants that differ only in the term used to describe the object that HyPE must generate:

1. Meta-prompt A: "instructive prompt"

2. Meta-prompt B: "prompt"

3. Meta-prompt C: "hypothetical prompt"

4. Meta-prompt D: "hypothetical instructive prompt" (our final formulation)

Each variant was evaluated under identical conditions across all benchmarks, using gpt-3.5-turbo with temperature 0.7. The results (Table 6) show that the precise wording of the target artifact substantially influences optimization quality. Although performance fluctuates across individual datasets, Meta-prompt D achieves the best overall average, suggesting that the combined specification of hypothetical and instructive leads to more faithful and more stable prompt optimizations.

Table 6: GPT-based HyPE optimization results comparison between meta-prompts. **Bold** values indicate results that outperform others.

| Dataset | Metric | Meta-prompt A | Meta-prompt B | Meta-prompt C | Meta-prompt D |
|---------|--------|---------------|---------------|---------------|---------------|
| SQuAD_2 | BertScore | 0.837 | **0.866** | 0.819 | 0.817 |
| GSM8K | EM | 0.23 | 0.19 | 0.27 | **0.75** |
| CommonGen | BertScore | 0.783 | **0.796** | 0.785 | 0.785 |
| AG News | F1 | 0.736 | **0.87** | 0.847 | 0.802 |
| XSum | BertScore | **0.718** | 0.717 | 0.71 | 0.715 |
| Mean result | | 0.661 | 0.688 | 0.686 | **0.774** |

### A.2.4 PROMPTS TRANSFERABILTY

To explore whether optimized prompts remain effective when applied in different contexts, we ran an additional set of transfer experiments. First, note that the XSum benchmark inherently spans a wide range of domains and topical distributions, including news, politics, sports, technology, and culture. Because of this built-in domain diversity, improvements obtained on XSum already serve as a useful indicator of cross-domain transferability: the same optimized prompt must perform well across heterogeneous content types rather than a single narrow domain.

Second, to evaluate cross-model transfer, we performed a controlled experiment on XSum using a simple manually written base prompt ("Summarize the sentence."). We optimized this prompt with HyPE using Qwen3-4B-Instruct-2507 as the target model, and then evaluated both the base and the optimized prompts on a set of 30 XSum examples across several different LLMs. The optimized prompt is shown in Figure 8, and Table 7 summarizes the corresponding evaluation results.

> You are a highly accurate and concise summarization assistant. Your task is to reduce a given sentence to its most essential meaning while preserving the core information, key entities, and intended message. Do not add, omit, or rephrase details beyond what is necessary to convey the original meaning. Maintain grammatical correctness and clarity. If the sentence is already concise, return it unchanged. Focus on brevity without sacrificing accuracy.
>
> Input sentence: {sentence}
>
> Output: A short, clear summary of the input sentence that captures its main idea in one or two grammatically correct sentences.

Figure 8: HyPE-optimized prompt for XSum

Although the prompt was optimized on a relatively small model, the improved version shows comparable or improved performance on several larger and architecturally different LLMs. This indicates that HyPE-optimized prompts can maintain their usefulness when transferred across models.

Table 7: Performance of HyPE-optimized prompt on XSum across different models. **Bold** values indicate results that outperform others.

| Model | Initial BertScore | Final BertScore |
|---|---|---|
| Qwen3-4B-Instruct-2507 | 0.6665 | **0.67** |
| mistral-7b-instruct | 0.487 | **0.508** |
| gemini-2.5-flash-lite | 0.596 | **0.706** |
| llama-3-8b-instruct | 0.66 | **0.68** |

## A.3 REFLECTIVEPROMPT

### A.3.1 EXPERIMENTS

ReflectivePrompt was compared with four other evolutionary-based methods of autoprompting: EvoPrompt, SPELL, PromptBreeder and Plum. For this comparison we used the following datasets: MNLI (Williams et al., 2018), MR (Chatterjee et al., 2021), SST-2 (Socher et al., 2013), YA-HOO (Kucuktunc et al., 2012), BBH (Suzgun et al., 2022), SamSUM (Gliwa et al., 2019).

All the computations were held on t-lite-instruct-0.1 and gemma3-27b-it models, and the results are shown in Tables 8-9.

Table 8: ReflectivePrompt and counterparts metrics on t-lite-instruct-0.1. BBH benchmark was divided into classification tasks group (a subset of datasets with strictly formatted answers that can be treated as classification task) and generation tasks group (dyck_languages, multistep_arithmetic_two, object_counting, word_sorting). The final BBH metrics (for both classification and generation groups) are the arithmetic mean of the metrics obtained for each dataset of the group separately. **Bold** values indicate results that outperform others.

| Dataset | Metric | EvoPrompt | SPELL | PromptBreeder | Plum | ReflectivePrompt |
|---|---|---|---|---|---|---|
| MNLI | F1-score | 0.537 | 0.734 | 0.476 | 0.564 | **0.738** |
| MR | F1-score | 0.642 | 0.633 | 0.932 | 0.617 | **0.958** |
| SST-2 | F1-score | **0.959** | **0.959** | 0.939 | 0.627 | 0.953 |
| YAHOO | F1-score | 0.438 | 0.420 | 0.473 | 0.291 | **0.507** |
| BBH (cls) | F1-score | 0.374 | 0.323 | 0.340 | 0.258 | **0.399** |
| SamSUM | METEOR | **0.450** | 0.442 | 0.427 | 0.447 | **0.450** |
| BBH (gen) | METEOR | 0.218 | 0.214 | 0.179 | 0.239 | **0.319** |

Table 9: ReflectivePrompt and counterparts metrics on classification task on gemma3-27b-it. BBH benchmark was divided into classification tasks group (a subset of datasets with strictly formatted answers that can be treated as classification task) and generation tasks group (dyck_languages, multistep_arithmetic_two, object_counting, word_sorting). The final BBH metrics (for both classification and generation groups) are the arithmetic mean of the metrics obtained for each dataset of the group separately. **Bold** values indicate results that outperform others.

| Dataset | Metric | EvoPrompt | SPELL | PromptBreeder | Plum | ReflectivePrompt |
|---|---|---|---|---|---|---|
| MNLI | F1-score | 0.597 | **0.602** | 0.582 | 0.587 | 0.599 |
| MR | F1-score | 0.642 | **0.958** | 0.956 | 0.637 | **0.958** |
| SST-2 | F1-score | 0.641 | 0.951 | 0.636 | **0.962** | 0.956 |
| YAHOO | F1-score | 0.635 | 0.627 | 0.590 | 0.615 | **0.636** |
| BBH (cls) | F1-score | 0.604 | 0.552 | 0.528 | 0.522 | **0.610** |
| SamSUM | METEOR | 0.423 | 0.425 | 0.406 | 0.423 | **0.426** |
| BBH (gen) | METEOR | 0.453 | **0.502** | 0.316 | 0.329 | 0.491 |

### A.3.2 IMPLEMENTATION DETAILS

## A.4 DISTILLPROMPT

### A.4.1 EXPERIMENTS

The experimental results across metrics and datasets are presented in Table 10 for the t-lite-instruct-0.1 model. The comparison was held between non-gradient autoprompting methods, such as Protegi and Grips. The datasets that were used for comparison are: SST-2 (Socher et al., 2013), MNLI (Williams et al., 2018), TREC (Li & Roth, 2002; Hovy et al., 2001), MR (Chatterjee et al., 2021), MedQA (Jin et al., 2021), BBH (Suzgun et al., 2022).

Table 10: DistillPrompt and counterparts metrics on classification task. Samples for few-shot was randomly selected from the training dataset. BBH benchmark was divided into classification tasks group (a subset of datasets with strictly formatted answers that can be treated as classification task) and generation tasks group (dyck_languages, multistep_arithmetic_two, object_counting, word_sorting). Protegi was not evaluated on generation tasks as its methology is not adapted for this. **Bold** values indicate results that outperform others.

| Dataset | Metric | Baseline prompt | Few-shot: n=3 | Protegi | Grips | DistillPrompt |
|---------|--------|-----------------|---------------|---------|-------|---------------|
| SST-2 | F1-score | 0.613 | 0.933 | 0.640 | 0.613 | **0.948** |
| MNLI | F1-score | 0.418 | 0.374 | 0.496 | 0.741 | **0.761** |
| TREC | F1-score | 0.287 | 0.268 | **0.355** | 0.315 | 0.353 |
| MR | F1-score | 0.862 | 0.603 | 0.636 | 0.912 | **0.939** |
| MedQA | F1-score | 0.296 | 0.240 | 0.293 | **0.303** | 0.296 |
| BBH (cls) | F1-score | 0.205 | 0.313 | 0.372 | 0.288 | **0.404** |
| SamSUM | METEOR | 0.448 | 0.385 | - | 0.455 | **0.458** |
| BBH (gen) | METEOR | 0.125 | 0.210 | - | 0.149 | **0.296** |

## A.5 KEY FEATURES OF PROPOSED FRAMEWORK

### A.5.1 INTERACTION WITH LLMS

CoolPrompt supports comprehensive LLM integration, ranging from locally deployed open-source models to proprietary API-based solutions. For standardized LLM interfacing, we implemented LangChain due to its provider-agnostic architecture that abstracts model-specific implementations, optimization techniques, and API variations. This design constitutes a critical framework component that democratizes LLM selection for end-users while eliminating the need for custom interface adaptation, a notable limitation present in alternative prompt optimization libraries.

### A.5.2 PROMPT IMPROVEMENT FEEDBACK

Beyond prompt optimization capabilities, CoolPrompt enhances methodological transparency by providing users with constructive feedback containing actionable suggestions and composition insights. This functionality is implemented through the PromptAssistant module, which performs a comparative analysis between initial and optimized prompt versions. PromptAssistant generates an interpretation of prompt optimization results, thereby contributing to the development of users' technical proficiency in prompt engineering and to exploration of "efficient prompt pattern".

### A.5.3 SYNTHETIC DATA GENERATOR

Modern LLMs have demonstrated remarkable efficacy in the resolution of instructional tasks, allowing the generation of synthetic data complete with target annotations. CoolPrompt takes advantage of this capability to address critical bottlenecks in prompt evaluation.

You are an expert in LLM task domain.
You are given a user's prompt.
Write the detailed problem description for which that prompt was created.
Use only textual description. Do not add another data.
Prompt: prompt
Provide your answer in JSON format with object with key "problem_description".
Output format:
{
    "problem_description": "Determined problem description"
}

Figure 9: Meta-prompt for problem description generating

You are an expert in synthetic data generation. You are very experienced in creating task examples.
You should create a validation dataset of {num_samples} examples.
Create a set of ground-truth labels. Then make some test questions (inputs) that correlates with problem description and use created labels as the responses.
Try to make the answers distribution more random.
Problem description: {problem_description}
Provide your answer in JSON object with key "examples" containing a list of your artificial examples. Each example is an object with keys "input" and "output" which contain corresponding text.
Make sure to include all necessary data in "input" object. You must not add any other objects except "input" and "output".
Also remember that "input" and "output" are textual fields. If you have some answer choices for input - just concat them with input text into one string.
Output format is the JSON structure below:
{
    "examples": [
        {
            "input": "Textual input",
            "output": "Ground-truth label",
            "id": 1
        },
        ...
        {
            "input": "Textual input",
            "output": "Ground-truth label",
            "id": {num_samples}
        }
    ]
}
Output JSON data only. Remember to create exactly {num_samples} examples.

Figure 10: Meta-prompt for classification task synthetic data generating

You are an expert in synthetic data generation. You are very experienced in creating task examples.
You should create a validation dataset of {num_samples} examples.
Create example pairs input-output that will correspond given problem description.
Problem description: {problem_description}
Provide your answer in JSON object with key "examples" containing a list of your artificial examples. Each example is an object with keys "input" and "output" which contain corresponding text.
Make sure to include all necessary data in "input" object.
You must not add any other objects except "input" and "output". Also remember that "input" and "output" are textual fields.
Output format is the JSON structure below: {
   "examples": [
     {
       "input": "Textual input",
       "output": "Correct model output",
       "id": 1
     },
     ...
     {
       "input": "Textual input",
       "output": "Correct model output",
       "id": {num_samples}
     }
   ]
}
Output JSON data only. Remember to create exactly {num_samples} examples.

Figure 11: Meta-prompt for generation task synthetic data generating

The generation process comprises three sequential phases.

1. Highlighting the initial problem description based on the prompt provided by the user. See meta-prompt in Fig. 9

2. Core synthetic data generation.

   - The meta prompt for this step may vary depending on the provided problem (either it is a classification or generation task). This variability is due to the need for more structured label generation for a future dataset in the case of a classification task. Meta-prompts are shown in Figures 10 - 11.

   - The use of ids in the dataset makes makes the generation more stable, since, depending on the quality of the model, LLM is not always able to correctly stop and generate a right number of examples.

3. Dataset expansion with hypothetical edge cases and complex scenarios incorporating. Meta-prompts for corner-cases scenarios generation are shown in Fig. 12 - 13.

You are an expert in synthetic data generation. You are very experienced in creating task examples.

You should create a validation dataset of {num_samples} examples.

Create a set of ground-truth labels. Then make some test questions (inputs) that correlates with problem description and use created labels as the responses. Try to make the answers distribution more random.

The key point of your task is to create as most corner and edge cases for the problem as possible. Try to think out of line to create the most difficult or non-trivial or corner scenarios you can imagine.

Your examples should not be easy in terms of guessing the right answer. They should be diverse and challenging.

Problem description: {problem_description}

Provide your answer in JSON object with key "examples" containing a list of your artificial corner-case examples. Each example is an object with keys "input" and "output" which contain corresponding text.

Make sure to include all necessary data in "input" object. You must not add any other objects except "input" and "output".

Also remember that "input" and "output" are textual fields. If you have some answer choices for input - just concat them with input text into one string.

Output format is the JSON structure below: {
    "examples": [
        {
            "input": "Textual corner-case input",
            "output": "Ground-truth label",
            "id": 1
        },
        ...
        {
            "input": "Textual corner-case input",
            "output": "Ground-truth label",
            "id": {num_samples}
        }
    ]
}

Output JSON data only. Remember to create exactly {num_samples} examples.

Figure 12: Meta-prompt for generating corner case examples for classification task

```
You are an expert in synthetic data generation. You are very experienced in creating
task examples.
You should create a validation dataset of {num_samples} examples.
Create example pairs input-output that will correspond given problem description.
The key point of your task is to create as most corner and edge cases for the problem
as possible. Try to think out of line to create the most difficult or non-trivial or corner
scenarios you can imagine.
Your examples should not be easy in terms of guessing the right answer. They should
be diverse and challenging.
Problem description: {problem_description}
Provide your answer in JSON object with key "examples" containing a list of your
artificial corner-case examples. Each example is an object with keys "input" and
"output" which contain corresponding text.
Make sure to include all necessary data in "input" object. You must not add any other
objects except "input" and "output".
Also remember that "input" and "output" are textual fields. If you have some answer
choices for input - just concat them with input text into one string.
Output format is the JSON structure below: {
    "examples": [
        {
            "input": "Textual corner-case input",
            "output": "Correct model output",
            "id": 1
        },
        ...
        {
            "input": "Textual corner-case input",
            "output": "Correct model output",
            "id": {num_samples}
        }
    ]
}
Output JSON data only. Remember to create exactly {num_samples} examples.
```

Figure 13: Meta-prompt for generating corner case examples for generation task

The ablation study on synthetic data quality is provided in the Appendix A.2.1.

### A.5.4    TASK DETECTOR

Task Detector is a specialized component or module designed to work in conjunction with LLMs. Its primary function is to analyze the input of a user prompt and automatically identify the intent and the specific type of task the user wants the LLM to perform.

Instead of manual setup by user, LLM, which could be as target LLM, as PromptAssistant, identifies a task problem that uses for specifying a target metric.

### A.5.5    COMPUTATIONAL COST ANALYSIS

In Table 11 you can see the cost comparison between three of our proposed methods with gpt-3.5-turbo model. The total cost is measured in USD. The execution time of the optimization methods in seconds is reported in Table 12.

This comparison, combined with our previous results and experiments, provides determinateness in selecting an optimization algorithm within our featured CoolPrompt framework. As shown in the results, ReflectivePrompt is our strongest method in terms of quality and performance. However, it is substantially more costly than the alternatives, as it requires a considerable number of model calls and has a longer runtime. DistillPrompt is a slightly more cost-effective option that also shows good

Table 11: Cost Comparison: each number in this table represents the average value per one epoch across five considered datasets ("AG News", "Common Gen", "SQUADv2", "XSum" and "GSM8k").

| Method | API calls | Prompt tokens | Completion tokens | Total tokens | Total cost (USD) |
|---|---|---|---|---|---|
| DistillPrompt | 409 | 77852.84 | 23567.76 | 101420.6 | 0.07431 |
| HyPE | 1 | 244.4 | 37.8 | 282.2 | 0.00018 |
| ReflectivePrompt | 487.2 | 152436.25 | 37647.75 | 190084.0 | 0.13269 |

Table 12: Execution Time Comparison (in seconds). For DistillPrompt and ReflectivePrompt, the metrics were calculated across 5 epochs.

| Method | SQUADv2 | Common Gen | AG News | XSum | GSM8k | Average | Average per epoch |
|---|---|---|---|---|---|---|---|
| DistillPrompt | 305 | 207 | 155 | 563 | 1026 | 451.2 | 90.24 |
| HyPE | 1.1 | 0.7 | 1.2 | 1 | 1.1 | 1.02 | 1.02 |
| ReflectivePrompt | 1077.33 | 1077.88 | 1078.03 | 1132.95 | 1087.46 | 1090.73 | 218.15 |

results in various domains. In cases where inexpensive and quick optimization of the user prompt is needed, HyPE represents an excellent choice, as it utilizes only a single LLM call to obtain results, being very fast and typically completing in about one second.

