# OpenReview forum: "CoolPrompt: An Automatic Prompt Optimization Framework for Large Language Models"
_ICLR.cc/2026/Conference — Submitted to ICLR 2026_

### Official Review · Reviewer_Z8Xp · 2025-10-31

**Soundness:** 2
**Presentation:** 3
**Contribution:** 1
**Rating:** 2
**Confidence:** 2

**Summary:**

This paper presents CoolPrompt, an automatic prompt optimization framework designed to be "zero-configuration". It automates the full optimization pipeline, including automatic task and metric selection, and synthetic data generation for when annotations are missing. The framework introduces three new optimization algorithms: HyPE (a rapid meta-prompting method),

**Strengths:**

1. The primary strength is the "zero-configuration" framework that aims to automate the entire autoprompting pipeline. This includes valuable components like a Task Detector, Synthetic Data Generator, and a PromptAssistant for optimization feedback, addressing significant practical barriers for users.

**Weaknesses:**

1. The evaluation is limited which doesn't benchmark against other prompt optimization methods such as OPRO.
2. I reserves conservation on the novelty of framework.
3. The proposed three prompt optimization approaches is not thoroughly benchmarked and analyzed if they are the main contribution.

**Questions:**

None

---

### Official Review · Reviewer_qvU3 · 2025-11-01

**Soundness:** 3
**Presentation:** 2
**Contribution:** 2
**Rating:** 4
**Confidence:** 4

**Summary:**

The paper introduces CoolPrompt, a zero-configuration framework for automatic prompt optimization that combines (1) a rapid meta-prompting method HyPE, and (2) two longer-running autoprompting algorithms ReflectivePrompt and DistillPrompt. The system also includes automatic task/metric detection, an LLM-driven synthetic data generator, and a PromptAssistant that provides feedback; experimental comparisons across several benchmarks show competitive results versus other autoprompting libraries.

**Strengths:**

1. The paper demonstrates an end-to-end system perspective by presenting a complete pipeline that includes task detection, synthetic data generation, optimization strategies, and feedback mechanisms, making it valuable as an applied system.
2. It introduces diverse optimization strategies—HyPE for meta-prompting, ReflectivePrompt for evolutionary reflection, and DistillPrompt for iterative distillation—that complement each other and are well motivated.
3. The work incorporates several practical features, such as model-agnostic integration through LangChain, automatic metric selection, and feedback generation, which strengthen its engineering contributions.
4. The authors conduct thorough ablations on synthetic data and meta-prompt design, evaluating synthetic data quality across models and refining the HyPE meta-prompt, making these analyses appropriate and informative.

**Weaknesses:**

1. The experimental setup is weakly grounded — each task uses only 30 samples and three runs, which is insufficient for statistically reliable conclusions.
2. The paper does not report standard deviations, confidence intervals, or significance tests, making it difficult to assess robustness. The optimization experiments rely on only 30 samples per task and three runs, which is too small to draw statistically significant conclusions. No confidence intervals, variance estimates, or error bars are provided.
3. Several baseline results appear inconsistent or potentially misconfigured (e.g., extremely low BertScore values), raising questions about fairness and reproducibility.
4. The paper lacks human evaluation for generative tasks and does not discuss known failure cases or limitations. All evaluations rely on automated metrics like BertScore or EM, which are insufficient for tasks involving coherence, relevance, or factual consistency. A small-scale human study would have strengthened the evidence.
5. Conceptual novelty is modest; the work focuses more on system integration than on algorithmic advancement.
6. Key configuration details are missing - including random seeds, prompt templates for baselines, number of LLM calls, runtime, and cost estimates. Without these, reproducibility is difficult, and results cannot be independently verified.Key configuration details are missing — including random seeds, prompt templates for baselines, number of LLM calls, runtime, and cost estimates. Without these, reproducibility is difficult, and results cannot be independently verified.
7. The paper does not analyze cases where automatic prompt optimization fails (e.g., tasks with ambiguous metrics or open-ended generation), nor does it provide insights into when human-in-the-loop guidance remains necessary.

**Questions:**

1. How many optimization trials and LLM calls were performed per method, and what were the compute costs?
2. How consistent are results when the same optimization is repeated with different random seeds?
3. How were baselines (Promptify, DSPy, etc.) configured, and were identical datasets, metrics, and generation parameters used?
4. What explains anomalies such as the very low BertScore for some baselines in Table 1?
5. How is circularity avoided when using GPT-4o or GPT-3.5 both as optimizer and as evaluation model?
6. What mechanisms ensure that automatically generated synthetic data are accurate and not hallucinated or mislabeled?
7. Can the authors demonstrate transferability?  Do optimized prompts generalize to unseen LLMs or unseen domains?
8. How does the proposed system handle tasks where evaluation metrics cannot be automatically inferred (e.g., subjective or multi-objective tasks)?

---

### Official Review · Reviewer_5tTe · 2025-11-02

**Soundness:** 2
**Presentation:** 3
**Contribution:** 2
**Rating:** 2
**Confidence:** 5

**Summary:**

This paper introduces a new prompt optimization framework, CoolPrompt, that receives a generic description of the task, and then uses a series of optimization processes to iteratively refine the prompt using real data, synthetic data, and LLMs. Three central techniques are used: (1) using an LLM to generate new prompts, (2) a genetic algorithm to extend and synthesize prompts, and (3) and an iterative generate and distill approach. The approach is tested on five type of tasks, with prompts from the second technique attaining higher performance on four of five

**Strengths:**

- Presents multiple options for how to optimize prompts automatically. Methods uses a series of steps that can each potentially contribute to the optimization

- Minor, but I agree with the authors that this paper is filling a key gap in prompt optimization for smaller open-weight models

- Compares performance on five types of tasks

**Weaknesses:**

- The paper describes a single framework but three approaches are analyzed (Table 2) and it is not clear to me whether the framework picks one of these methods or if their prompts are somehow aggregated. Given the superior performance of the ReflectivePrompt setup, it's not clear why the other approaches are needed (perhaps they do better for some settings?)

- The whole pipeline is relatively complex, which isn't necessarily bad. However, it's not clear what in this pipeline is contributing to a better prompt. Some type of ablation analysis would be very helpful here beyond what's in A.2. It would help to see what impact each stage has on the resulting performance.

- While the models are getting better performance with the prompt, I would be very interested in the timing of the pipeline, given its complexity. Compared to other approaches, how much longer (or shorter) is CoolPrompt?

- The tasks, while diverse, are relatively old with AG News being over a decade. I'm not opposed to older tasks if they still pose a challenge but given the age, most pretrained/large language models have seen this data which adds a potential confound. Also, given the relatively high performance on all tasks, even for the manual zero-shot prompt, it would be useful to see whether these approaches work for more recent and more challenging tasks.

- Minor, but many experiment details are moved to the appendix. however, the paper has nearly a page of extra space. It would make the paper much more readable to have this content in the main part.

**Questions:**

- I was confused by the comment on line 251 that CoT requires task-specific exemplars. I don't think this is true since you could just include the "Think step by step" command in the prompt to elicit CoT output. Could you clarify what is meant here?

---

### Official Review · Reviewer_cWFZ · 2025-11-02

**Soundness:** 2
**Presentation:** 1
**Contribution:** 1
**Rating:** 2
**Confidence:** 5

**Summary:**

This paper presents CoolPrompt, an automatic prompt optimization framework for large language models. The framework provides a zero-configuration workflow including automatic task detection, metric selection, synthetic data generation, and optimization feedback. The authors propose three optimization methods: HyPE for rapid optimization, and ReflectivePrompt and DistillPrompt for long-term optimization. Experiments on multiple benchmark datasets demonstrate the framework's effectiveness.

**Strengths:**

1. This work proposes a relatively complete prompt optimization framework with commendable attention to data aspects (including automatic data generation and task detection), which has significant practical value for real-world applications.
2. The work introduces several optimization algorithms synthesized from existing approaches (HyPE, ReflectivePrompt, DistillPrompt) and demonstrates improved performance over baselines across multiple tasks.

**Weaknesses:**

1. The paper's presentation requires significant improvement. Figures 1, 2, and 3 use crude flowcharts that rely heavily on text stacking, lacking clear visual hierarchy, making it difficult for readers to quickly grasp key information and inter-module relationships.
2. While claiming to present a complete framework, the paper omits critical implementation details. Particularly, the synthetic data generation module, listed as a core contribution, is only described with a four-step outline (Appendix A.5.3) without providing specific generation algorithms, prompt templates, or quality control mechanisms, making this contribution difficult to verify and reproduce.
3. The experimental design has significant flaws. First, the benchmark datasets are dated and the experimental scale is too small (only 30 samples total, with just 6 samples in the training set). Second, the comparison targets are inappropriate, only comparing against engineering frameworks like Promptify and AdalFlow, rather than mainstream optimization algorithms such as OPRO, DSPy, and APE. Finally, the experiments lack statistical significance testing, ablation studies, and cost analysis.
4. The paper lacks substantial academic novelty. ReflectivePrompt and DistillPrompt are essentially combinations of existing evolutionary algorithms and Tree-of-Thoughts methods. While HyPE shows some originality, it lacks in-depth analysis. The synthetic data generation is not a novel method but merely an engineering integration. Overall, this reads more like a technical report for an engineering project rather than an academic paper with significant methodological contributions.

**Questions:**

Please provide detailed implementations of key modules, particularly the complete algorithmic workflow and data quality control mechanisms for synthetic data generation.

---

### Official Review · Reviewer_w4An · 2025-11-03

**Soundness:** 2
**Presentation:** 2
**Contribution:** 2
**Rating:** 2
**Confidence:** 4

**Summary:**

This paper proposes an automatic prompt optimization system for large language models (LLMs) that includes automatic task and metric selection, and synthetic data generation. The proposed method provides three prompt optimization strategies: ReflectivePrompt, DistillPrompt, and HyPE. The effectiveness of the proposed method is verified through experiments on several tasks, including question answering, mathematical reasoning, and text classification.

**Strengths:**

- This paper provides a zero-configuration framework for prompt optimization, which is useful for users who are not familiar with prompt engineering.
- In the proposed method, three different prompt optimization strategies are provided for short- and long-term optimization.
- The experimental results demonstrate that the proposed method exhibits competitive performance on different tasks with existing prompt optimization methods.

**Weaknesses:**

- Although this paper proposes a complete pipeline for prompt optimization, each component of the proposed method is based on existing methods. For instance, the prompt optimizers, HyPE and ReflectivePrompt, are based on prior works of HyDE (Gao et al., 2023) and Reflective Evolution (Ye et al., 2024). The technical novelty of the proposed method is limited, and the core innovation of this paper is not clear.
- The experimental evaluation is weak:
    - Only the gpt-3.5-turbo model is considered. It is unclear whether the proposed method works well for other LLMs.
    - Only the performance of each prompt optimization method is evaluated. However, the cost of each method in terms of LLM API calls or token consumption is not discussed.

**Questions:**

- How is the computational cost of each prompt optimization method in terms of LLM API calls or token consumption?
- The authors claim that other automatic prompt optimization frameworks have limitations of usage only for proprietary LLMs. However, the reviewer cannot understand the reason for this point. What are the technical limitations and difficulties of existing methods when using custom LLMs?

---

### Author Response · Authors · 2025-11-19
**Official Comment**

Thank you for your reviews.

To Reviewer w4An:

About key novelty of proposed methods:
The key novelty of the ReflectivePrompt is the application of the reflective evolution method within the framework of the autoprompting task (originally it was used as a part of Language Hyper-Heuristics for different COP solutions and was not considered as a prompt optimization technique).
While HyPE is inspired by the hypothetical-generation idea in HyDE, our contribution is distinct: we apply this paradigm to prompt optimization rather than retrieval. HyPE provides a data-free, single-step enhancement procedure that avoids task-specific exemplars and iterative search, using only one LLM call to produce a refined, task-aligned instruction.
About cost analysis: Check Appendix 5.5.
About the technical limitations of existing methods when using custom LLMs:
Since current frameworks have a specific list of available models, the limitation is that using a custom model requires additional time and development expertise to adapt the solution's source code. If modified incorrectly, this can also affect the optimization results. Furthermore, accessing models via an API is not the only method available. Our library implements other common native approaches for running local models.

To Reviewer cWFZ:
We provided detailed implementation information in Appendix A 5.3.

About key novelty of proposed methods:
Check answer 1 to Reviewer w4An


To Reviewer cWFZ and qvU3:
Reminder for the experimental setup:
The main settings for the automatic optimization libraries were set to default. Identical initial prompts, data for the optimization process, and model generation parameters were used for the experiment.
Clarification regarding the datasets: The reason for using this small optimization dataset size is to demonstrate the effectiveness of the prompt optimization solution with limited data. The data for optimization was randomly selected and identical for each framework, respectively.

To Reviewer 5tTe:

Clarification about CoT:
In this section we were referring to the original formulation of Chain-of-Thought (CoT) prompting introduced by Wei et al. (2022)
Justification for several proposed methods and computational cost analysis: Check Appendix 5.5.
About relatively old datasets: Noted, thank you, we will keep this in mind for our future research.
About pipeline contribution: The primary contribution to prompt optimization comes from the optimization algorithms themselves (HyPE, DistillPrompt, ReflectivePrompt). Modules such as TaskDetector, Synthetic Data Generator, and others are used to address engineering challenges such as implicit metric or task type parameters, lack of labeled data, and other issues. Therefore, in this case, we can discuss the effectiveness and competitiveness of the automatic prompt optimization algorithms, which are sufficiently described in our paper.

To reviewer qvU3:

Computational cost and execution time comparison is shown in Appendix 5.5
Indeed, such an anomaly exists for this dataset: the optimized prompt from Promptify failed to produce summarization results, which consequently caused the BERTScore value to drop.
About LLM Circularity:
LLM circularity does not arise in our experiments because:
1. We rely primarily on non-LLM evaluation metrics.
2. When using an LLM-based metric (e.g., G-Eval), we separate the target and system models.
About synthetic data generator: Appendix A 5.3.
About prompts transferability: please see Appendix 2.4.
Our framework is capable of handling complex tasks for two reasons:
1. Support for custom, user-defined evaluation functions (including LLM-based ones such as G-Eval).
2. The benchmark suite already covers diverse and nontrivial task types in our experiments. The datasets used in our evaluation span several fundamentally different task families.

To ICLR 2026 Conference Program Chairs and Senior Area Chairs:
I am writing to formally appeal the decision and assessment of the review for our submission. While we respect the rigorous peer-review process, we have identified several substantive issues in the provided review that we believe warrant your reconsideration. A rejection based on a lack of novelty requires identifying the specific literature that preempts these contributions, which the reviewer has not done. The review states the evaluation is "limited" and does not benchmark against "other prompt optimization methods such as OPRO." It does not specify which aspects of our comprehensive evaluation (e.g., on diverse tasks, with synthetic data) are considered "limited." Given these points, we believe the review does not provide a fair or sufficiently expert assessment of our work. We respectfully request that you seek an additional, independent review from a reviewer with demonstrated expertise in automatic prompt optimization to ensure a balanced evaluation. Thank you.

---

### Meta-Review · Area_Chair_U6J1 · 2026-01-11

**Summary:**

The paper proposes **CoolPrompt**, a “zero-configuration” framework intended to automate an end-to-end prompt optimization workflow for LLMs. The system includes modules for **automatic task detection**, **automatic metric selection**, **dataset splitting and/or synthetic data generation when labels are missing**, and a feedback/reporting component (“PromptAssistant”). Within this framework, the paper introduces three prompt optimization strategies: **HyPE** (a single-step meta-prompting refinement approach), and two longer-running optimizers, **ReflectivePrompt** and **DistillPrompt**. Experiments across several tasks suggest CoolPrompt can achieve competitive performance relative to selected baselines, with additional analyses/ablations (noted by reviewers) on aspects such as synthetic data and meta-prompt design.

### Strengths
- **End-to-end, practical pipeline**
### Weaknesses
- **Limited methodological novelty**: Multiple reviewers (w4An, cWFZ, qvU3, Z8Xp) argue that key components are heavily inspired by or composed from existing ideas, and that the paper’s core algorithmic innovation is not clearly articulated.
- **Evaluation weaknesses and missing rigor**: Several reviewers raise concerns about small experimental scale (e.g., ~30 samples/task), lack of variance reporting/significance testing, and limited benchmarking against widely-used optimization algorithms (cWFZ, qvU3, Z8Xp). Some baselines appear potentially misconfigured or anomalous (qvU3).

This submission has received overlapping, substantive concerns about **evaluation rigor, reproducibility/verification, and limited novelty**. While the rebuttal addresses some points (notably pointing to cost analysis and providing clarifications), it does not resolve the central soundness and experimental-design objections raised by several reviewers.

**Reviewer Concerns:**

Concerns substantially addressed by the rebuttal (at least partially):
- **Cost/runtime question**: Authors point to a cost/execution-time comparison in Appendix 5.5 (raised by w4An, 5tTe, qvU3). This is a step toward addressing the request, though several reviewers also asked for clearer reporting/standardization in the main text and method-by-method accounting.
- **Baseline configuration fairness (claimed defaults + same prompts/data/params)**: Authors state baselines used default settings and identical initial prompts/data/parameters, and acknowledge an anomaly for one summarization baseline (qvU3 concern). This addresses the question at a high level, though it does not fully resolve reproducibility concerns without concrete configs/prompts/seeds.
- **LLM circularity**: Authors claim circularity is avoided by primarily using non-LLM metrics and by separating target vs evaluation/system models when using an LLM-based metric (qvU3 concern).

Concerns still outstanding after the rebuttal:
- **Insufficient experimental rigor (sample size, uncertainty, statistical testing)**: The rebuttal does not convincingly resolve concerns about the small scale and lack of standard deviations/CIs/significance tests (cWFZ, qvU3).
- **Benchmarking scope and baseline selection**: Concerns about missing comparisons to commonly cited optimization algorithms (e.g., OPRO mentioned by cWFZ and Z8Xp) remain; the rebuttal does not indicate added experiments.
- **Synthetic data generation reproducibility and quality control**: While authors point to Appendix A 5.3, reviewers’ core concern was the lack of specific algorithms/prompt templates/QC mechanisms sufficient for reproduction (cWFZ, qvU3). The rebuttal does not clearly add those missing details.

**Reviewer Scores:**

I don't expect any changes.

---

### Decision · Program_Chairs · 2026-01-26

Reject